# Assessing the Connectivity of Urban Green Spaces for Enhanced Environmental Justice and Ecosystem Service Flow: A Study of Tehran Using Graph Theory and Least-Cost Analysis

**Seyed Ali Alavi** [1,†], **Saeed Esfandi** [2,3,†], **Amir Reza Khavarian-Garmsir** [4], **Safiyeh Tayebi** [5,6,*],
**Aliakbar Shamsipour** [7] and **Ayyoob Sharifi** [8,9]

1   Geographic Information System Analyst, Tehran Sewerage Company, Tehran 15875-5696, Iran; a.alavi@tsc.ir
2   Center for Energy and Environmental Policy, Joseph R. Biden, Jr. School of Public Policy and Administration,
    University of Delaware, Newark, DE 19716, USA; sesfandi@udel.edu
3   Foundation for Renewable Energy and Environment (FREE), New York City, NY 10111, USA
4   Department of Geography and Urban Planning, Faculty of Geographical Sciences and Planning,
    University of Isfahan, Isfahan 8174673441, Iran; a.khavarian@geo.ui.ac.ir
5   Department of Geography, Rutgers the State University of New Jersey, New Brunswick, NJ 08854, USA7
6   Rutgers Global Health Institute, Rutgers the State University of New Jersey, New Brunswick, NJ 08901, USA
7   Faculty of Geography, University of Tehran, Tehran 1417853933, Iran; shamsipr@ut.ac.ir
8   The IDEC Institute & Network for Education and Research on Peace and Sustainability (NERPS),
    Hiroshima University, 1-5-1 Kagamiyama, Higashi Hiroshima City 739-8529, Hiroshima, Japan;
    sharifi@hiroshima-u.ac.jp
9   School of Architecture and Design, Lebanese American University, Beirut P.O. Box 13-5053, Lebanon
*   Correspondence: safiyeh.tayebi@rutgers.edu
†   These authors contributed equally to this work.

**Abstract:** This research aims to analyze the relationship between environmental justice and urban green space connectivity in Tehran, Iran. The evaluation of green space connectivity in this study is conducted through two distinct cost layers: one aimed at enhancing existing connections and another focused on establishing new green spaces. Key factors influencing connectivity, extracted from the relevant literature, were identified to facilitate this analysis. Employing graph theory and least-cost analysis, the results determined critical resistance factors, current connectivity and cost status, the varying degrees to which different districts benefit from green space ecosystem services, and the most effective routes for establishing green corridors. Research findings highlight significant disparities in access to these services, particularly in the underserved central districts of Tehran. Moreover, spatial analysis reveals a higher potential for enhancing east–west ecosystem service corridors due to the higher density of green hubs and lower costs in this orientation, while north–south connectivity faces more challenges. By exploring land use/land cover, and physical and socio-economic factors affecting urban green space connectivity, this study provides urban and environmental planners with a novel methodology and comprehensive insights for effective decision making, resource allocation, and land use planning.

**Keywords:** urban ecosystem services; urban green spaces; connectivity; spatial inequality; environmental justice; Tehran

## 1. Introduction

Rapid urbanization has led to the urban population rising from 0.8 billion in 1950 to 4.4 billion in 2020, and it is projected to reach 6.7 billion by 2050. This increase puts even more pressure on cities to develop housing, services, and infrastructure to meet this growth [1,2]. Such urban expansion, often driven by the conversion of natural landscapes into urban developments, not only places additional pressure on urban carrying capacity [3], but also exacerbates environmental injustice and inequality [4–6]. In response to these

urbanization challenges and in line with nature-based solutions, urban green infrastructure (UGI) has emerged as a promising strategy to mitigate the adverse environmental impacts of rapid urban growth [7]. UGI integrates green spaces into the urban fabric, offering a multifaceted approach to combating issues such as greenhouse gas emissions, urban heat islands, and stormwater runoff, while simultaneously enhancing socioeconomic and public health benefits [8–12].

Central to the concept of UGI is the notion of connectivity [13], which involves creating integrated networks of green spaces and corridors, bridging the divide between natural and human landscapes within cities [14]. This multifunctional approach prioritizes the efficient use of urban land and the equitable provision of urban services to citizens [15]. UGI often consists of various green spaces, water bodies, green corridors, valleys, and channels, where larger and more closely connected green spaces and corridors contribute to better environmental performance and higher ecological efficiency [16].

The literature on green space connectivity can be categorized into two primary strands. The first strand investigates the structural connectivity of natural landscapes, concentrating on the causes of landscape fragmentation and its repercussions on biodiversity [17–20]. Researchers in this area employ a variety of network analysis and evaluation methods such as graph theory to evaluate and enhance structural connectivity [16,21]. These studies offer critical insights into the structural attributes of green space networks. The second strand delves into functional connectivity and the movement of ecosystem services within urban settings. These studies use spatial analyses to assess different factors to optimize ecosystem services and minimize pressure on various urban resources. They apply techniques such as satellite imagery and spatial correlations to model green infrastructure connectivity in urban areas [7,22–25]. While structural connectivity focuses on the physical relationships between green spaces, functional connectivity is pivotal as an operational parameter influencing the flow of ecosystem services, setting the stage for a deeper analysis within this context [26,27].

On the other hand, recent scholarly discourse surrounding urban green spaces increasingly acknowledges their significant role in determining environmental justice outcomes. For instance, Enssle and Kabisch [28] and Wolch et al. [29] argued that the distribution and connectivity of green spaces are critical factors in facilitating equitable access to environmental benefits. Calderón-Argelich et al. [30] underscored the necessity of a comprehensive framework for urban ecosystem service assessments, with a particular focus on the equitable distribution of UGI. Meanwhile, Zuniga-Teran and Gerlak [31] advocated for a multidisciplinary approach to justice issues in urban green spaces, emphasizing the importance of aspects such as funding, safety, connectivity, multifunctionality, and community engagement.

The literature review highlighted a notable deficiency in studies addressing how connectivity patterns influence the flow of ecosystem services and, in turn, affect environmental justice in urban settings. Furthermore, while previous research has extensively explored the physical aspects of accessibility to green spaces and the movement of ecosystem services, it has often neglected the integral role of socioeconomic and demographic factors [32]. This oversight is particularly apparent in the limited exploration of the interplay between these factors and both ecosystem services and urban green spaces [33]. In addition, urban greening projects tend to overlook the social dimensions of urban green spaces [30], typically prioritizing other aspects and services [34]. Despite existing research offering valuable insights into Tehran's environmental justice concerning access to green spaces [35] and ecosystem services [32], there remains a lack of explicit focus on how green space connectivity directly impacts environmental justice. Crucially, a comprehensive analysis is needed to fully understand the implications of these connectivity patterns on environmental justice within urban contexts, especially considering the multifaceted nature of urban green space connectivity.

Reflecting on the multifaceted nature of urban green space connectivity, it is crucial to recognize that it involves not just enhancing physical linkages but also augmenting the flow of ecosystem services. Ecosystem services, as defined by Seppelt et al. [36], are the

diverse benefits humans derive from ecosystems, including social, ecological, and economic advantages, and intangible benefits such as cultural, recreational, and health aspects. This comprehensive understanding is essential to ensure equitable access for all urban residents, regardless of socioeconomic background [28,37]. However, in densely populated cities such as Tehran, the concept of 'green space' becomes more context-dependent, changing significantly based on urban dynamics [38]. Furthermore, the spatial distribution of green spaces is heavily influenced by a range of land use and socioeconomic factors, reflecting the complex interplay between urban development and environmental planning [39]. Given the challenges of fair access in today's compact and densely populated cities, the proposed solution is the structural and functional connectivity of urban green spaces [40].

Building on this solution and the definitions provided for connectivity and ecosystem service concepts, this research posits that enhancing the connectivity of urban green spaces through the development of structural and functional corridors can facilitate equitable access to ecosystem services for all urban districts and neighborhoods in Tehran. In fact, this study aims to augment the existing body of knowledge by implementing a nuanced analysis tailored to the unique physical, socioeconomic, and demographic dynamics of Tehran in the context of urban green space connectivity. An integrated graph theory and least-cost analysis method was developed, incorporating both structural and functional perspectives. With its comprehensive and adaptable framework, this method is generalizable and can be effectively applied to other urban settings to assess and enhance green space networks and ecosystem services. By adjusting key parameters to reflect local ecological, socioeconomic, and urban land use characteristics, this research offers valuable insights and practical implications for urban planners and policymakers. Research findings facilitate informed decision making and improve urban resource allocation, fostering environmental justice and equitable access to green spaces and ecosystem services.

## 2. Study Area

Tehran, the capital city of Iran, with a population exceeding 9 million residents across 353 neighborhoods and 22 municipal districts (Figure 1), ranks as the 38th most populous city in the world [41]. Centralized economic and social policies, coupled with an absence of territorial planning for the equitable distribution of services, facilities, and population in Iran, have led to vast spatial inequalities nationwide. This has led to large-scale migration to Tehran, and the emergence of less-developed peripheral areas and delipidated urban fabrics in central districts. Tehran has experienced significant land use and population density changes, particularly over the past five decades. These changes have predominantly occurred in Tehran's western and southern parts, as physical constraints in the north and east limit expansion and land cover modifications [6].

The average green space per capita in Tehran's neighborhoods is approximately 16.5 square meters, but there is a significant variance between neighborhoods. The highest per capita is 291 square meters, while the lowest is virtually zero [42]. This stark contrast is especially pronounced in ten neighborhoods in central and southern Tehran, where urban density is three times the city's average and devoid of parks and green spaces. This disparity underscores the inequity in green space distribution and highlights a lack of spatial justice. Figure 1 illustrates the main green spaces (over 11 hectares) and population density of Tehran, revealing that districts with high population density have limited access to these green spaces in terms of proximity.

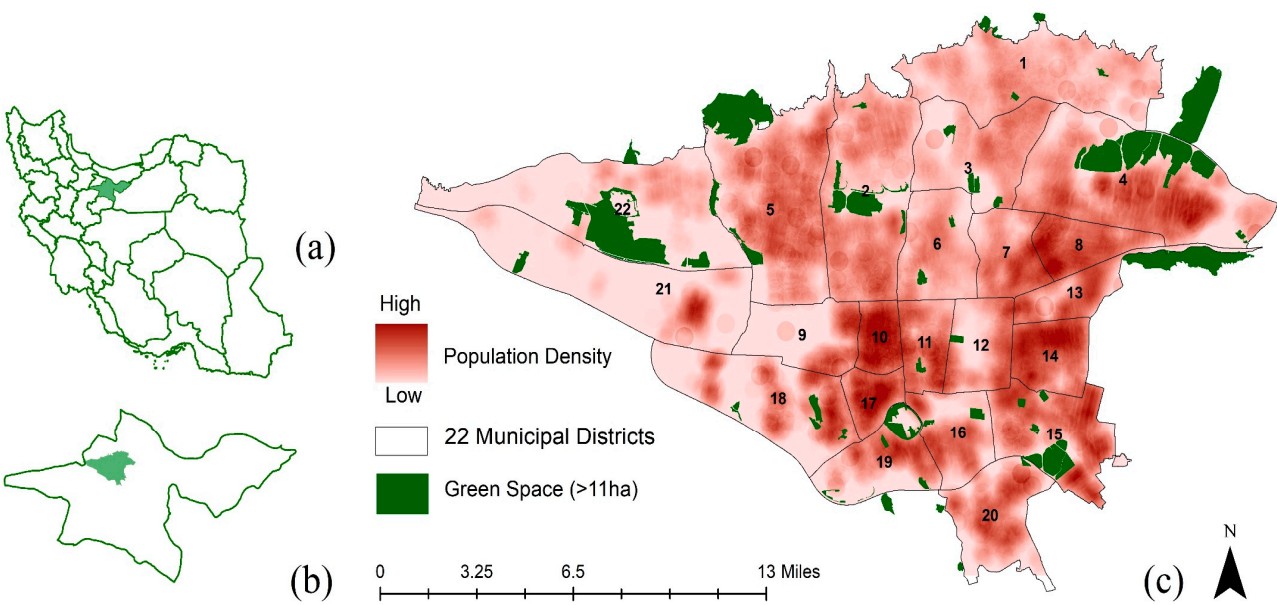

**Figure 1.** (**a**) Tehran Province location in Iran, (**b**) Tehran city location in Tehran Province, and (**c**) population density and urban green space distribution in Tehran.

## 3. Materials and Methods

The primary objective of this study is to evaluate the connectivity of urban green spaces and the flow of ecosystem services in Tehran, with a particular focus on urban environmental justice. This evaluation is designed to estimate the extent to which urban residents benefit from the ecosystem services provided by green spaces. The relationship between connectivity and the utilization of ecosystem services is complex and nonlinear, influenced by various urban social and environmental factors [43]. To analyze connectivity, we used graph theory and the least-cost model, with the assumption that 100% structural connection represents an ideal scenario. Our assessment of connectivity included considering several factors increasing or decreasing resistance to (1) the flow of ecosystem services between existing green spaces, and (2) developing new green spaces. As a practical tool for urban governance, our study presents a network model to enhance connectivity and pinpoint areas that require targeted intervention. As outlined in Figure 2, this research was conducted in five steps.

### 3.1. Determining Factors Affecting Green Space Connectivity

To identify the factors influencing the connectivity of green spaces and the flow of ecosystem services, we commenced our study with a comprehensive review of the existing literature. This review culminated in the identification of 17 key factors pertinent to assessing the resistance to ecosystem service flow among existing green spaces. Additionally, two demographic factors were incorporated into evaluating resistance in the context of creating new green spaces, as elaborated in Table 1. To quantitatively assess the influence of these factors, an online questionnaire was administered, supplemented by the Delphi method. Initially, 47 experts were identified based on their familiarity with the concepts presented in the paper and their knowledge of Tehran. This identification was primarily based on their previous research and work experience in relevant fields. The selected experts included professionals and academics from various disciplines such as urban planning, climatology, urban geography, ecology, environmental design, agriculture, and landscape architecture. The questionnaire was then sent to these experts, out of whom 35 agreed to participate. The panel comprised an international group of experts, bringing diverse perspectives and expertise to the study.

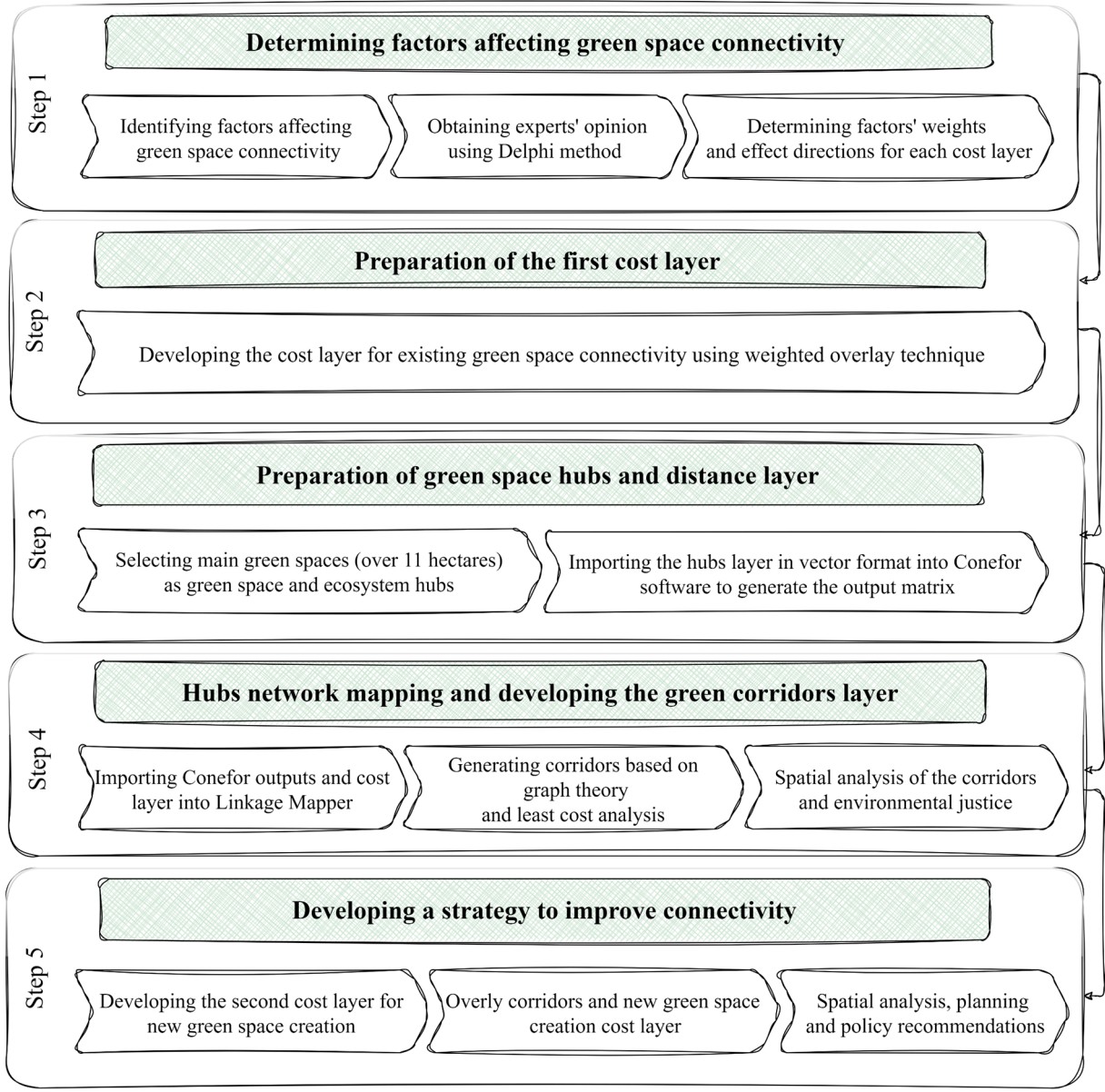

**Figure 2.** Flowchart of the research steps.

These experts evaluated the relevance and intensity of the impact of these factors on a dual scale of 1 to 10, once considering the creation of new green spaces and once in the context of ecosystem service flow among existing green spaces. The derived weights from this evaluation were subsequently normalized to a scale ranging from 0 to 1, enabling precise spatial weighting in our analysis. It should be noted that as the Delphi method is inherently iterative, after obtaining the initial weights from the first round of expert opinions, we sent the results back to the panel for a second review. This enabled the experts to re-evaluate their responses in light of the collective feedback, ensuring a more refined and consensus-based outcome. The final weights, achieved through this iterative process, were then used to inform our spatial weighting and analysis, adding robustness and depth to our study's findings.

**Table 1.** Factors and their assigned weights.

| Factor | | Resistance to Creating New Green Space | | Resistance to the Flow of Ecosystem Services between Existing Green Spaces | | Source |
|---|---|---|---|---|---|---|
| | | Effect Direction | Normalized Weight | Effect Direction | Normalized Weight | |
| UGI factors (over 11 hectares) | Distance between green spaces | ↓ | 0.089 | ↑ | 0.091 | [16,44,45] |
| | Area of green spaces | ↑ | 0.083 | ↓ | 0.085 | |
| | Distance of urban parcels from green space | ↓ | 0.031 | ↑ | 0.032 | |
| Land use/Land cover factors | Residential | ↑ | 0.074 | ↑ | 0.076 | [32,35,45–49] |
| | Other green spaces (below 11 hectares) | ↑ | 0.061 | ↓ | 0.062 | |
| | Urban Services | ↑ | 0.038 | ↑ | 0.039 | |
| | Commercial | ↑ | 0.038 | ↑ | 0.041 | |
| | Military | ↓ | 0.054 | ↑ | 0.051 | |
| | Agricultural and garden | ↓ | 0.053 | ↓ | 0.052 | |
| | Urban facilities and equipment | ↑ | 0.038 | ↑ | 0.052 | |
| | Mixed residential–commercial | ↑ | 0.051 | ↑ | 0.061 | |
| | Mixed residential–urban service | ↑ | 0.038 | ↑ | 0.054 | |
| | Water bodies | ↑ | 0.038 | ↓ | 0.041 | |
| | Barren land | ↓ | 0.047 | ↓ | 0.057 | |
| | Building density | ↑ | 0.079 | ↑ | 0.076 | |
| Socioeconomic and demographic factors | Population density | ↓ | 0.054 | ↑ | 0.054 | [12,16,21,38,45,50,51] |
| | Land price | ↑ | 0.086 | ↑ | 0.075 | |
| | The ratio of the population aged under 14 to the total population | ↓ | 0.025 | - | - | |
| | The ratio of the population aged over 60 to the total population | ↓ | 0.022 | - | - | |

↓ The higher value (for UGI and socioeconomic factors) and existence (for land use/land cover factors) decrease resistance. ↑ The higher value (for UGI and socioeconomic factors) and existence (for land use/land cover factors) increase resistance.

### 3.2. Preparation of the First Cost Layer

This step involved the collection and spatial analysis of data relevant to the identified factors influencing existing green space connectivity. To this end, we developed a cost layer, representing the cumulative resistance to the flow of ecosystem services. This layer was constructed by modeling the 17 identified factors through spatial analysis methods, particularly focusing on the weighted overlay approach. This technique allowed for a nuanced assessment of the connectivity among existing green spaces, providing a critical foundation for further analysis and strategic urban environmental planning.

### 3.3. Preparation of Green Space Hubs and Distance Layer

A critical step in establishing connections between green spaces was the creation of a network layer based on the graph model. A matrix reflecting the positions and distances of the green spaces was prepared by inputting the layer of 57 green spaces into Conefor software (Version 2.6) in vector format. Conefor software, which applies graph theory, is utilized in conservation programs due to its ability to calculate the habitat availability index. It quantifies the importance of areas and habitat connections for maintaining or improving landscape connectivity, evaluates the impact of habitat and land use changes on connectivity, and supports decision making in landscape conservation planning by identifying critical locations for ecological connectivity [44,52]. The selection of the 57 green spaces in this study aligns with the criteria set forth in Tehran's Green Spaces Master Plan. This plan mandates that, for every 51,000 residents, there must be a city-scale green space, spanning more than 11 hectares. These designated green spaces, which function as crucial hubs for ecosystem services in our analysis, are distinguished by their unique spatial characteristics and the density of various tree species. This strategic selection underscores the significance of these spaces in urban environmental planning and ecosystem service provision [42].

### 3.4. Developing the Corridors Layer

In this step, we utilized graph theory and least-cost analysis to establish connectivity between green spaces, referred to as hubs. The output matrix from Conefor software and the connectivity cost layer were utilized to draw connections and links using Linkage Mapper software (Version 2.0.0). Linkage Mapper is a tool designed to create ecological networks, calculating the distance between hubs based on the edge-to-edge Euclidean distance of adjacent hubs, thus offering a more realistic representation of distance and displacement. The results of this step include two distinct layers: active and inactive links or corridors. These outputs were then overlaid on the connectivity cost layer to analyze the green space connectivity and its barriers in Tehran. The least-cost method is instrumental in identifying the optimum routes between green spaces. This method evaluates the cost of a route, indicating its suitability for species or service movement [51]. In fact, least-cost modeling, grounded in graph theory algorithms, determines the most efficient corridors between green spaces by considering the cost layer [24].

### 3.5. Developing a Strategy to Improve Connectivity

Building on the spatial analysis of the corridors, this step aimed to evaluate the feasibility of creating new green spaces to enhance connectivity and promote environmental justice. A new cost layer was generated, designated as the cost layer for creating new green spaces. This layer, developed in consultation with experts from the Delphi panel, incorporates factors from the green space connectivity cost layer with different directions and intensity of impact, along with two demographic factors. The feasibility of establishing new green spaces in areas with weaker green space connectivity was assessed by analyzing the overlay of the new cost layer with the corridors layer. This analysis serves as a tool for urban strategic planning, focusing on improving green space accessibility in less privileged areas.

## 4. Results and Discussion

### 4.1. Resistance Factors

Regarding the flow of ecosystem services, the highest resistance is observed in factors such as 'Residential land use', 'Land price', and 'Population density', with residential land use and population density emerging as particularly significant barriers. This aligns with the findings of Davies et al. [46], Hepcan [47], and Schetke et al. [48], who similarly highlight the challenges posed by dense urban settlements. High land prices further exacerbate these challenges, as they often lead to prioritizing commercial and residential development over the creation or expansion of green spaces. These factors collectively hinder the seamless integration and connectivity of green spaces in urban landscapes. The situation underscores an urgent need for strategic land-use planning and policy making that balances urban development with environmental sustainability. Such policies should aim to incorporate green spaces more effectively within the urban fabric, recognizing their critical role in enhancing urban ecosystems, improving biodiversity, and providing essential services to urban populations.

In the context of creating new green spaces, the analysis underscores the pivotal role of 'Distance to green spaces', 'Land price', 'Area of green spaces', 'Building density', and 'Other green spaces' (below 11 hectares). As observed, the area of green spaces, with a high weight, indicates increased resistance to the creation of larger green spaces. This may be linked to logistical challenges and the higher costs of developing and maintaining larger areas. The variables 'Distance to Green Spaces' and the presence of 'Other Green Spaces (smaller than 11 hectares)' indicate that regions more remote from existing green spaces, or those devoid of such spaces, may encounter reduced resistance to the development of new green areas. This is contingent upon the conditions of diminished land-use conflicts, lower urban pressures, and more favorable land prices in these areas. Moreover, the high weight of 'Building density' resonates with the idea that densely built areas might oppose the creation of new green spaces due to existing infrastructural and land-use commitments.

The analysis also highlights social dimensions, particularly demographic factors such as the ratio of the population aged under 14 and over 60, in shaping demand for green space ecosystem services. These age groups, with their distinct needs for varied green spaces—active areas for the young and tranquil spots for the elderly—demonstrate how social aspects can shape the demand for ecosystem services. Conversely, the provision and nature of these ecosystem services also significantly impact the social fabric of urban communities [36]. As noted by Hegetschweiler et al. [53], understanding this demand-supply interplay is crucial. Although our study focused mainly on physical and demographic factors, limited by access to comprehensive social data, it points towards the necessity for future research to integrate broader social variables for effective urban green space planning. Such an integrative approach ensures that the diverse community needs are met, enhances urban social sustainability, and acknowledges the dynamic interplay between ecosystem services and social aspects.

Integrating these insights provides a comprehensive view of the challenges in creating new green spaces and maintaining ecosystem services flow in urban areas. It underlines the importance of considering a multitude of factors, including land use, building density, socioeconomic conditions, and community demands, in urban planning and policy-making for sustainable urban green space development.

*4.2. Existing Green Space Connectivity Cost Layer*

The application of weighted overlay and spatial analysis to the 17 identified factors culminated in the creation of a cost layer for existing green space connectivity. As illustrated in Figure 3, this layer delineates the varying levels of resistance to green space connectivity, with values ranging from 0.41 to 0.85. Notably, the central districts of the city, marked by high population density, extensive residential land use, and significant building density, exhibit the highest levels of resistance and associated costs. This pattern of resistance is also discernible in certain western and southern districts, primarily due to their relative remoteness from green spaces, which serve as ecosystem service hubs. Conversely, areas in close proximity to green spaces demonstrate the lowest resistance to the flow of ecosystem services, as indicated by the dark blue color on the map. The resistance progressively increases in regions further away from these hubs, a phenomenon that is significantly influenced by both the physical, socioeconomic and demographic attributes of these areas.

As Figure 3 illustrates, the central districts of Tehran (7, 9, 10, 11, 12, 14, and 17) demonstrate the lowest connectivity levels, which correlates with high building and population densities, and per capita green space averages below the city's overall figure, as reported by Tehran Municipality [42]. These districts also suffer from inadequate public transportation infrastructure and a noted deficiency in private vehicle ownership, exacerbating the challenges of accessing green spaces. Being the historical heart of the city and home to significant landmarks such as the Tehran Grand Bazaar, these areas pose unique challenges for urban renewal and green space allocation. Addressing environmental justice here necessitates innovative strategies for green space creation in such historical and underserved areas, as recommended by Calderón-Argelich et al. [30] and Chaudhary et al. [54]. Approaches such as urban land greening, repurposing transportation infrastructure, and nature-based solutions such as green roofs, walls, and pocket parks are advocated to enhance ecosystem service provision and improve urban resident well-being [29,55–58].

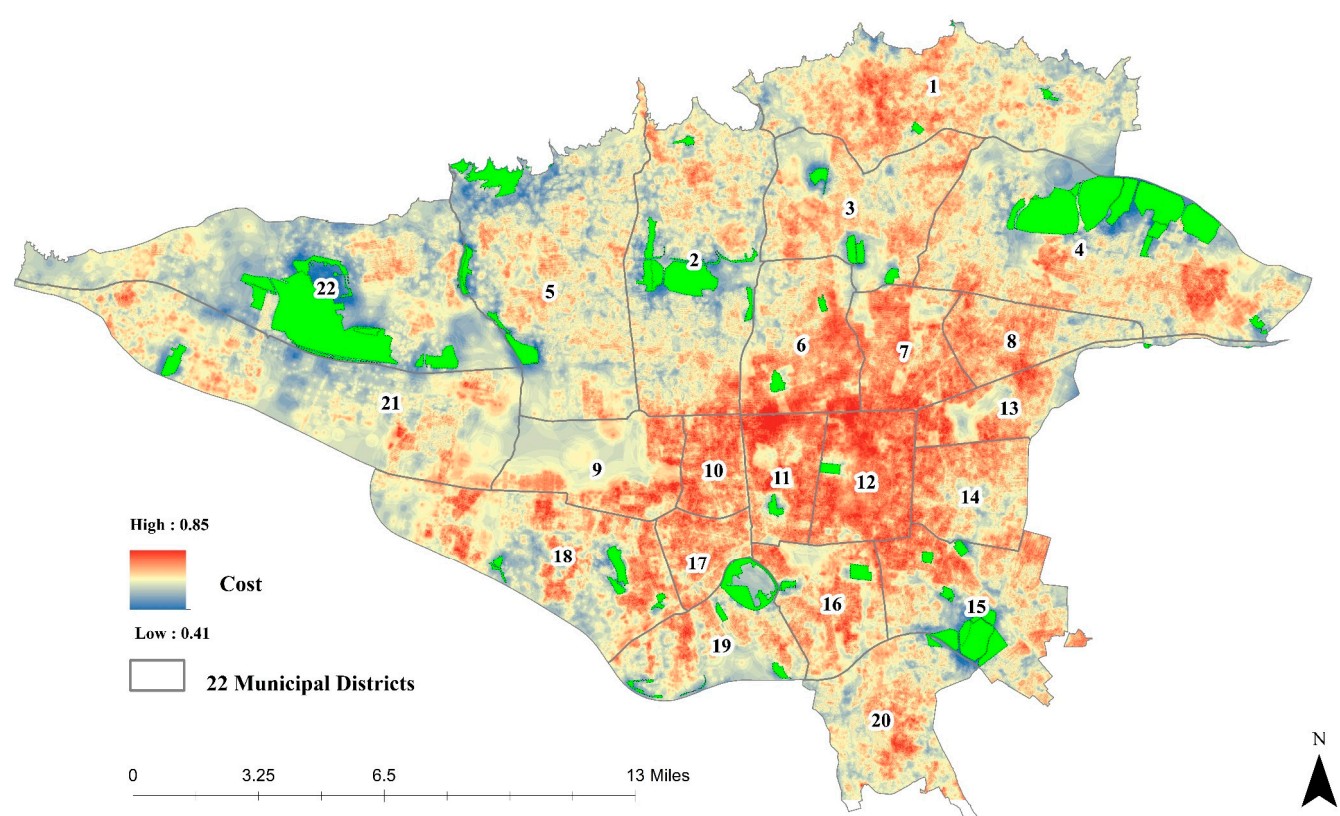

**Figure 3.** Existing green space connectivity cost map.

### 4.3. Hub Network Mapping and Developing the Corridor Layer

In accordance with our methodology, 57 green spaces were designated as ecosystem service hubs. Conefor software's graph model was employed to establish all potential networks connecting these hubs, as depicted in Figure 4a. Initially, these networks were formed without excluding any connections and assigned equal weights to represent the full range of possible linkages. This comprehensive network served as the foundational input for the Linkage Mapper software. Within Linkage Mapper, the integration of the cost layer with these networks facilitated the assignment of differentiated weights to each connection, as shown in Figure 4b.

Following this integration, a rigorous filtering process was applied to these networks. Through the application of least-cost analysis, we were able to delineate corridors, subsequently categorizing them based on their respective weights, illustrated in Figure 4c. The refinement provided by the least-cost analysis significantly enhanced the accuracy of corridor mapping, effectively incorporating the cost layer into the spatial analysis, as evidenced in Figure 4d. This approach not only ensured a more precise identification of green corridors but also allowed for a nuanced understanding of the connectivity dynamics among the identified green hubs.

The graph model, utilizing the cost layer and the green space network, categorized green corridors into four groups based on their connectivity levels (Figure 5). Corridors with the strongest connectivity, depicted in dark blue, are vital for the effective flow of ecosystem services. The most prevalent corridors, shown in yellow, demonstrate weaker connectivity, while those in red, particularly concentrated in the city center, indicate the lowest connectivity levels.

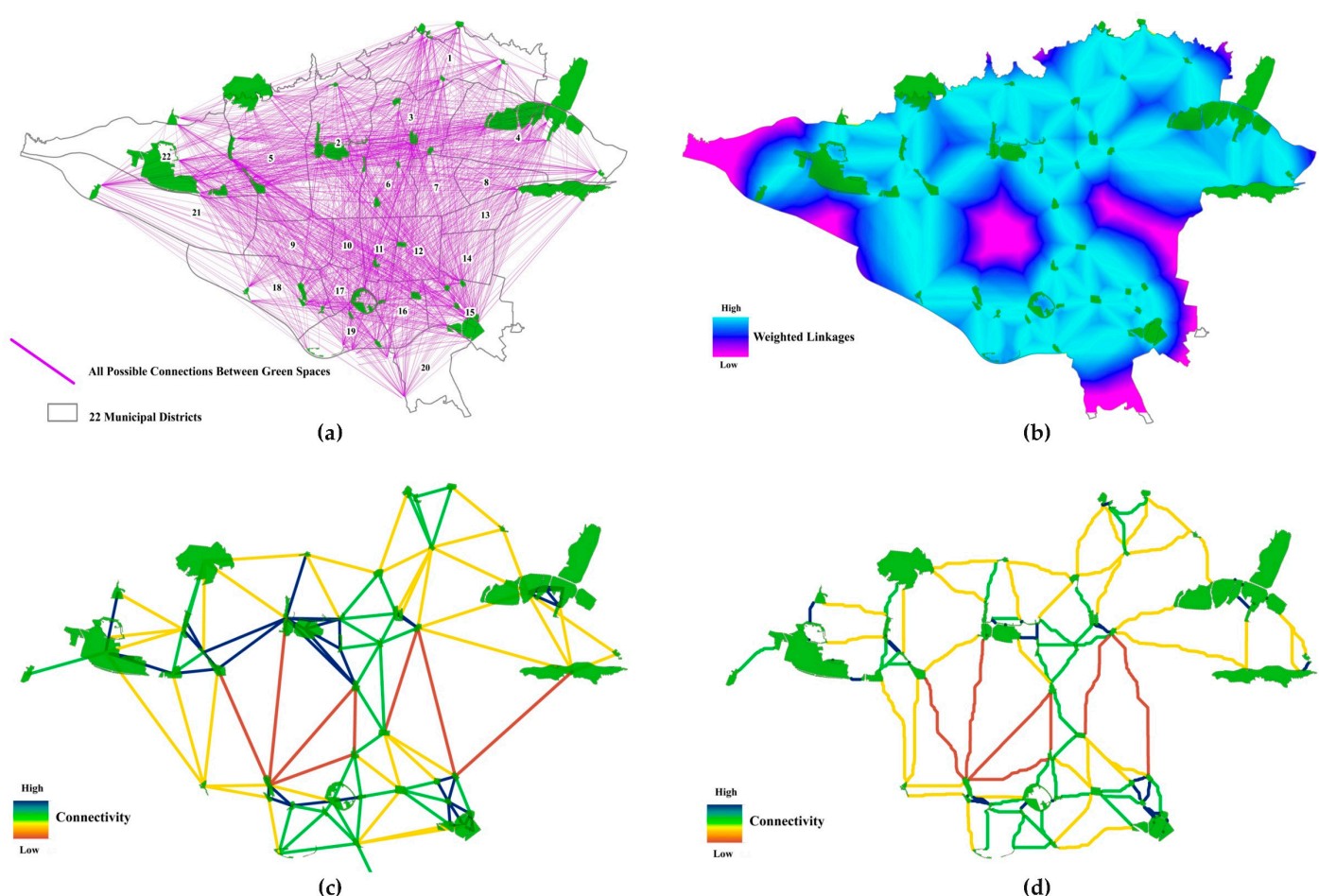

**Figure 4.** Connectivity analysis of green hubs. (**a**) initial network, (**b**) weighted linkages, (**c**) filtered corridors, (**d**) cost layer-informed corridors.

Figure 5 also reveals that corridors in the northern districts of Tehran (22, 5, 2, 6, and 3) display high connectivity, benefiting from large green spaces and close proximities between them. Some southern districts (18, 19, 15, and 16), with their open spaces, lower building density, and larger but fewer green spaces, similarly exhibit strong connectivity. A juxtaposition of the corridor map with Tehran's population distribution uncovers a substantial portion of the city's populace facing limited access to green space ecosystem services, predominantly due to poor functional connectivity within their vicinities. In 207 out of Tehran's 353 neighborhoods, many corridors are found to be suboptimal, impairing their role in ecosystem service delivery. Remarkably, these neighborhoods, housing approximately 6 million residents or about 75 percent of Tehran's total population, confront considerable challenges in accessing green space ecosystem services. Furthermore, 66 neighborhoods, collectively home to around 1.6 million people, are in an especially precarious situation regarding connectivity, marking the lowest availability of green space ecosystem services in this category.

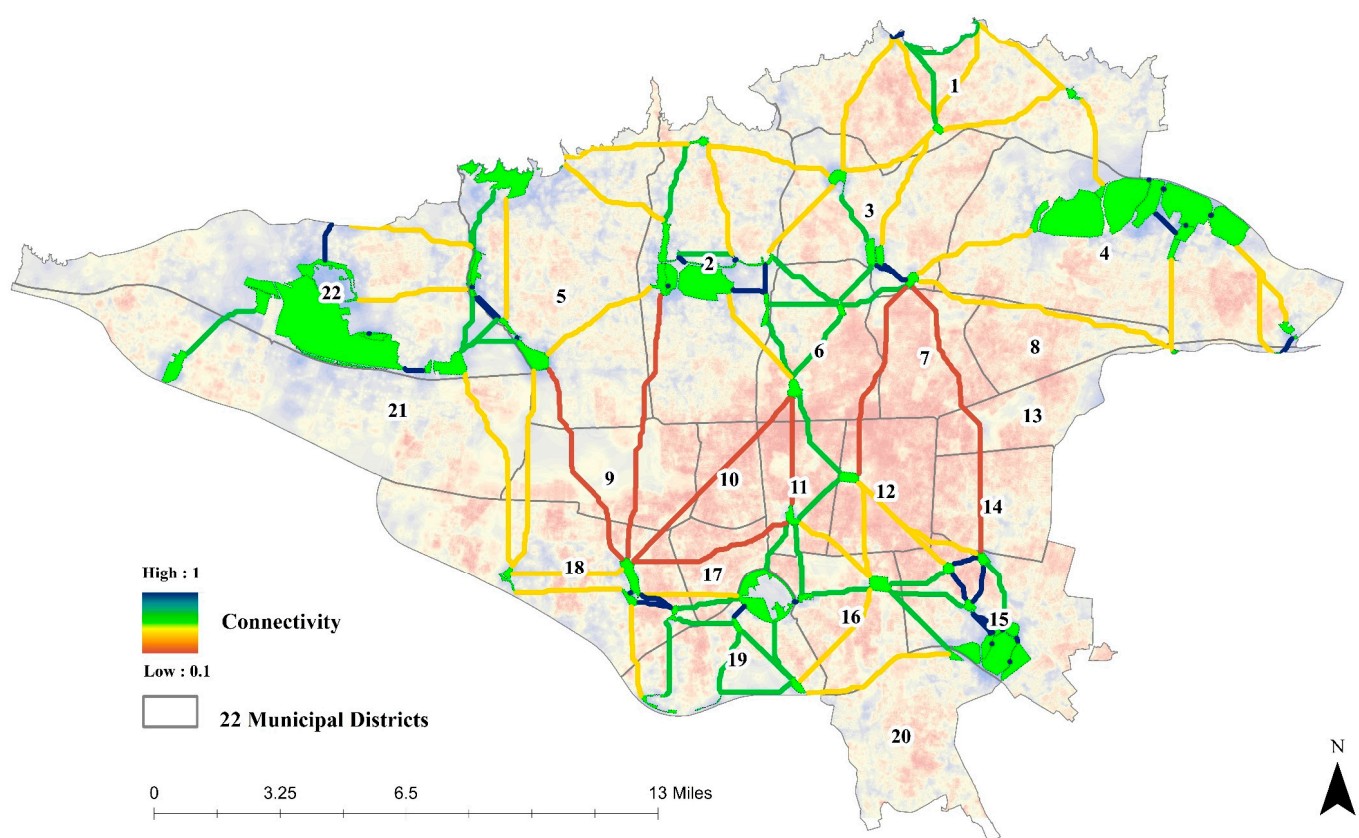

**Figure 5.** Corridors on the cost map of existing green space connectivity.

Figure 5 reveals two areas with the highest cost and red corridors, indicative of poor connectivity. These areas, as further illustrated in Figure 6, are marked by maximum building density and a paucity of urban green space, featuring the longest distances between green spaces within the city. Notably, these areas encompass the oldest districts of Tehran, namely districts 11, 12, and 10. The historical development of these districts demonstrates the imbalance between the expansion of built-up areas and the integration of green spaces over time, leading to the current situation. Despite some open spaces due to old urban textures and underutilized lands such as warehouses, the combination of long distances, high population, and building density hinders connectivity development, resulting in minimal access to ecosystem services for residents in these areas. Remarkably, some neighborhoods in these zones lack even a single square meter of green space.

The situation in Tehran is compounded by the absence of specific zoning regulations for high-rise buildings, which are predominantly situated in the western and northern districts. These structures impede natural ventilation, especially through west–east winds, a problem that is particularly pronounced in central areas. Here, the historical character of some neighborhoods has restricted the proliferation of high-rise buildings, leading to a 'suffocating hole' effect. Thus, the potential for neighborhood development might lie in reverse regeneration and the implementation of green strategies, as suggested by Ghasemi et al. [59], Rajaei and Mansourian [60], Reyhan et al. [61], Rezaei et al. [62], and Zargari et al. [11].

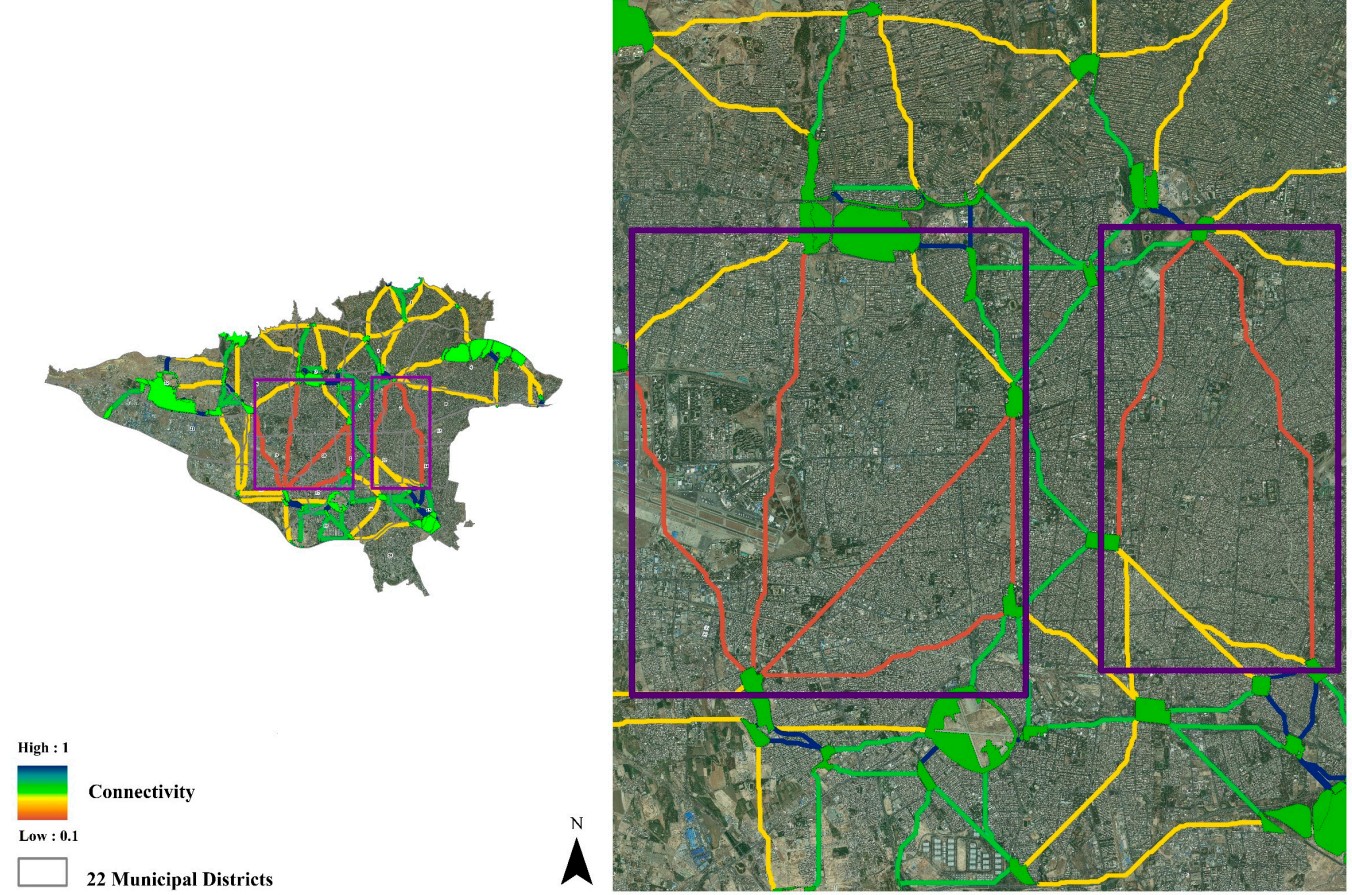

**Figure 6.** High-cost areas with limited green space connectivity in central Tehran.

In stark contrast to the aforementioned trend, there is a noticeable surge in construction within these districts. Despite prevalent criticism of housing developments, the dilapidated fabric, frequently considered prime for redevelopment into denser, multi-story housing, continues to fuel housing construction. This trend is not only driven by the higher land prices in central Tehran but also by the objectives of infill development. Nonetheless, this approach sharply diverges from the documented deterioration in the environmental quality of these neighborhoods. It appears that the focus is predominantly on maximizing land use for residential and other purposes, with green spaces receiving markedly less attention.

### 4.4. Developing a Strategy to Improve Connectivity through New Green Spaces

The analysis of connectivity corridors (shown in Figure 5) and their comparison with the existing green space connectivity cost layer indicate that east–west connections in the northern part of Tehran are most favorable. This is attributed to the denser concentration and east–west alignment of green hubs in this area. Conversely, corridors running in a north–south direction are weakly connected, mainly due to the lack of effective hubs along this axis. As previously discussed, these areas, marked by dense populations and an urgent need for urban renewal, should be key considerations in formulating land use plans and in determining building and population densities, particularly in the central parts of the city. Considering the high cost and scarcity of land in these regions, innovative solutions such as green roofs and other alternative green space forms would be beneficial.

Additionally, to support the direction and number of corridors and the need for new green spaces, a supplementary cost layer was developed, encompassing 19 factors influencing the creation of new green spaces (Figure 7). While these factors largely mirror those affecting existing green space connectivity, the direction and weight of their impact on resistance varies. For instance, increased distance between green spaces heightens

resistance to existing ecosystem services but lowers resistance to the establishment of new green spaces. Two social factors were also integrated into this layer to reflect local community needs for green spaces. The resistance values in this layer ranged from 0.23 to 0.68, with variations around the hubs based on their usage. The western parts of the city, characterized by less developed, agricultural, and barren lands, exhibit lower resistance to the development of new green spaces, whereas the central and eastern parts show high resistance.

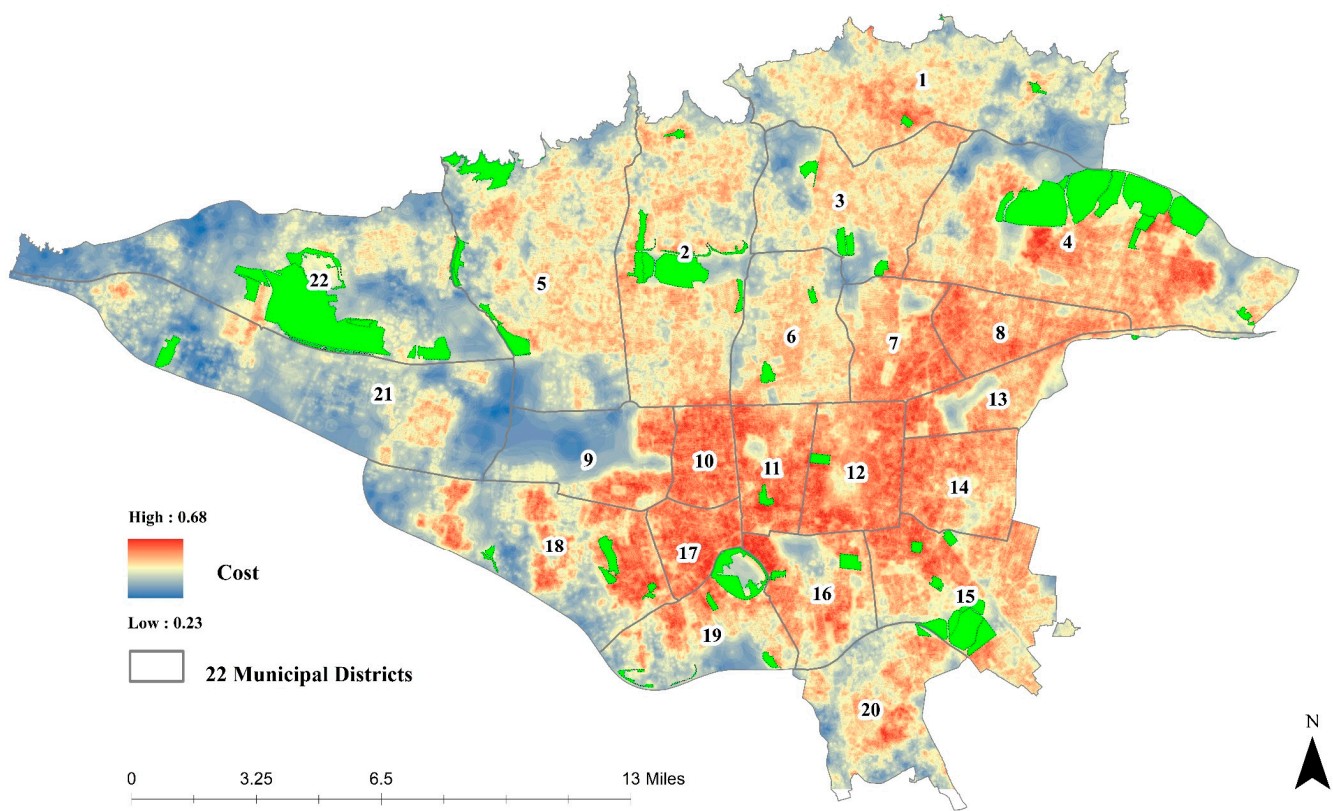

**Figure 7.** Cost map of developing new green spaces to improve connectivity and flow of ecosystem services.

As shown in Figure 8, while the western and northern parts of the city host more ecosystem service hubs and corridors, they exhibit a higher capacity for the development of new green spaces due to lower resistance and cost. Particularly in the western areas, where the potential for green space development is most pronounced, there exist substantial opportunities to expand green areas and ecosystem services in a cost-effective manner. However, this development may exacerbate the existing issues of uneven growth and spatial disparity in the distribution of green spaces throughout Tehran. Conversely, the presence of red corridors and areas in the cost layer for new green space creation highlights the considerable cost associated with establishing new hubs, especially in Tehran's central districts. This complex scenario underscores the need for innovative and adaptive connection strategies in future urban land use planning to address these challenges effectively.

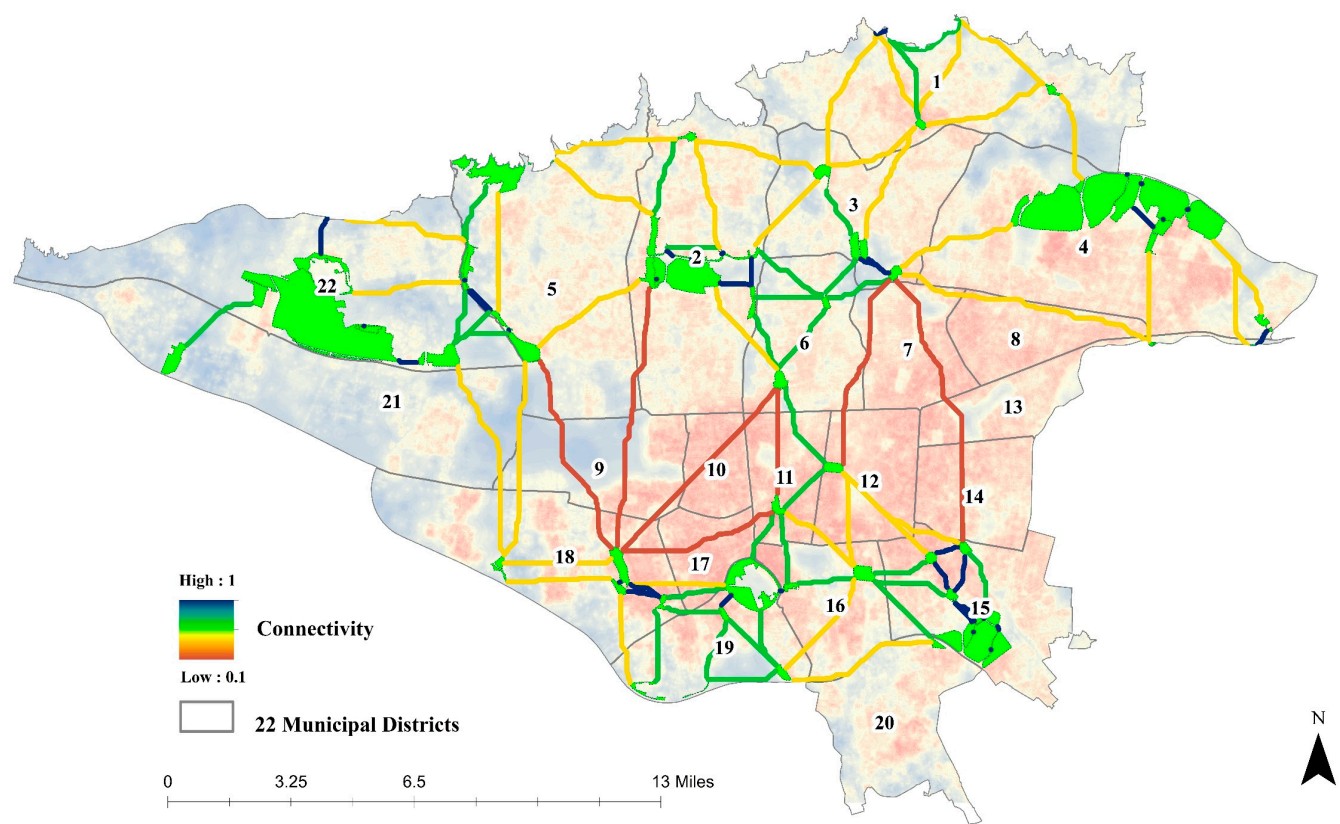

**Figure 8.** Corridors on the cost map of developing new green spaces.

The obtained results and findings, when compared to those of previous studies conducted by Xue et al. [63] and Kaushik et al. [64], suggest that implementing roof and wall greening strategies, along with incentivizing green space integration in private buildings, can significantly enhance both structural and functional connectivity. Several studies, including those by Kristianova and Vitkova [65], Yung et al. [66], and Iojă et al. [67], have underscored the potential of transforming dilapidated buildings into green public and open spaces. Additionally, they suggest the renovation of these structures with green roofs and walls as a means to enhance green space connectivity. Relevant research on Tehran's green space network [35,68] also underscores the need for urban management and land use policy interventions, particularly in Tehran's southern districts, to enhance per capita green space availability and access.

Our findings are in line with prior studies emphasizing the importance of environmental justice in urban landscape transformations and their community impacts. Echoing Zuniga-Teran and Gerlak's [31] concerns, our study identified the uneven distribution of green spaces, especially in wealthier neighborhoods, and highlights the need for collaborative, cross-disciplinary strategies to address these justice issues. This aligns with Wolch et al. [29], who observed the disproportionate benefits of green spaces in affluent communities, underscoring the significance of green space access as a key environmental justice matter. Moreover, our results are congruent with Jennings et al. [69], who noted the link between unequal access to urban green spaces and variables such as socioeconomic status. Together, these studies, along with our own, underscore the vital necessity of addressing the interplay between urban green space connectivity and environmental justice, reinforcing its significance in urban planning and policy formulation.

## 5. Conclusions

Urban growth exerts significant physical and spatial impacts, notably the loss of green spaces, which serve as vital remnants of nature within cities. These spaces undergo both quantitative and qualitative changes due to rapid urbanization, leading to ecological,

economic, and social consequences. Effectively planned green infrastructure can yield substantial social and economic benefits, forge connections between people and nature, provide private environmental spaces, and enhance ecosystem services when sustainable planning and design principles are applied. Land use planning, combined with sustainable urban design, can optimize the utilization of green spaces and their ecosystem services, thereby reducing urban costs and externalities, and enhancing city resilience.

This study revealed that Tehran's central districts, predominantly inhabited by more deprived social classes, have the least potential for developing corridors that connect ecosystem service hubs. Furthermore, the spatial analysis indicates a higher potential for ecosystem service flow in the east–west direction of the city, where ecosystem service hubs are more concentrated. In contrast, the north–south direction faces significant connectivity challenges.

The findings of this research are poised to inform the city's five-year green space planning and land use strategies. These strategies, including affordable housing, the renovation of deteriorated areas, and the regeneration of multifacetedly challenged neighborhoods, are in harmony with the intervention areas identified in our study. Addressing the diverse urban landscape of Tehran, our recommendations cater to different urban zones. For peripheral districts, where barren lands are more prevalent, we advocate for the development of a green belt. This approach aligns with initiatives started by the Tehran municipality a few years ago, which the authors suggest should be accelerated and expanded. In contrast, the densely populated and built-up central areas of the city, which face a scarcity of available land, would benefit more from urban forestry strategies. Implementations such as green walls and roofs are particularly apt for these regions. To facilitate this transition, we propose that the municipality should offer incentives and credits to property owners and developers who participate in these urban greening activities. This dual strategy, combining the expansion of peripheral green belts with the integration of urban forestry in central districts, offers a comprehensive and viable approach to enhancing Tehran's UGI.

Methodologically, this research introduces a new approach to land use planning studies by combining graph and least-cost theories to assess the flow of ecosystem services from green space hubs. This methodology, typically used in ecology, is novel in urban planning studies; it measures both structural and functional connectivity and analyzes physical and demographic factors through spatial analysis methods. The findings offer valuable insights for cities similar to Tehran in terms of socioeconomic conditions, geographical location, climate change effects, and environmental characteristics. This study contributes to the development of the knowledge network on ecosystem service values of green spaces in urban sciences, clarifying, modifying, and improving methodologies in urban planning.

Future research should explore the effectiveness of diverse green infrastructure strategies in different urban contexts, assessing their impact on ecosystem services, social interaction, and economic benefits. Investigating the social equity dimensions of green space access, particularly in economically disadvantaged areas, is crucial for promoting equitable distribution and accessibility. Integrating climate change adaptation strategies into green space planning using nature-based solutions is key for enhancing urban resilience. Comprehensive cost–benefit analyses of sustainable land use planning and green space development will provide valuable insights for decision makers and urban planners.

Despite these insights, it is important to recognize the limitations of this study. The methodologies employed, while innovative and globally applicable, may require further validation and refinement. The applicability of our findings to other cities should be considered with caution, and future research efforts could focus on further refining these methodologies.

**Author Contributions:** Conceptualization, S.A.A., S.E., A.R.K.-G. and A.S. (Ayyoob Sharifi); data curation, S.A.A., S.E. and S.T.; formal analysis, S.E., S.T., A.S. (Aliakbar Shamsipour) and A.S. (Ayyoob Sharifi); investigation, S.A.A., A.R.K.-G. and A.S. (Aliakbar Shamsipour); methodology, S.A.A., S.E., A.R.K.-G., S.T. and A.S. (Ayyoob Sharifi); project administration, S.A.A., A.S. (Aliakbar Shamsipour) and A.S. (Ayyoob Sharifi); Resources, S.A.A., S.E., S.T. and A.S. (Aliakbar Shamsipour); software, S.A.A., S.E., S.T. and A.S. (Aliakbar Shamsipour); supervision, S.A.A. and A.S. (Aliakbar Shamsipour); validation, A.R.K.-G., S.T., A.S. (Aliakbar Shamsipour) and A.S. (Ayyoob Sharifi); visualization, S.T.; writing—original draft, S.E. and S.T.; writing—review and editing, S.E., A.R.K.-G. and A.S. (Ayyoob Sharifi). All authors have read and agreed to the published version of the manuscript.

**Funding:** This research received no external funding.

**Data Availability Statement:** Data are contained within the article.

**Conflicts of Interest:** Author Seyed Ali Alavi was employed by the company Tehran Sewerage Company. The remaining authors declare no conflicts of interest.

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
