# Peer review of "Assessing the Connectivity of Urban Green Spaces for Enhanced Environmental Justice and Ecosystem Service Flow: A Study of Tehran Using Graph Theory and Least-Cost Analysis"

_urbansci, doi:10.3390/urbansci8010014_

Round 1

Reviewer 1 Report

Comments and Suggestions for Authors

The study analyses the existence of public urban green space and its structural connectivity and contrasts that data with a number of factors which determine the city's 'resistance' to create new [public] urban space, thus improving the provision of ecosystem services and contributing to increase life quality within the city of Tehran.

The manuscript is clearly structured and I detect no issues regarding its clarity and readability. Although I am not a native English speaker, I find it to be well written. The cited references seem also well chosen and seem sufficient and adequate to the discussion, although a few may still be added.

General suggestions:

The introduction is clear and well framed by the references used. My only comment is that authors should explain better in what way do the chosen factors (especially the socioeconomic and demographic factors) express a measure of "environmental justice", and how is the proposed analytical method capable of expressing its increase/decrease over time.

Also regarding the 'Introduction' and the 'Materials and Methods' sections, authors often mention 'green space' or 'urban green space', but what is seen in Figure 1 and throughout all of the study is that they only consider 'public green spaces', especially public parks and gardens. A quick view of Tehran's aerial photographic mosaic will reveal a finer 'mesh' of [private] urban green spaces that are left out of the study, but obviously contribute with ecosystem services and amenities to the city and are, in my perspective, not neglectable. Thus, authors should be clear(er) on how they chose the patches of [public] green spaces, and why was that their choice.

I am not familiar with the softwares used (neither with the methodology) but but the results and the discussion seem well conducted, except I would also like to see a diagram expressing the age of the different neighbourhoods, because it seems to me that a number of factors (population density, per capita urban space area, average building height) are highly correlated with the age of the neighbourhoods, within Tehran. There should be a significant amount of literature mentioning that this is a common problem to many 'historical' cities within the Eurasian continent.

Finally, authors "suggest a 'green belt plan' and the integration of vital ecosystem service corridors into the city's spatial organization". Do you consider that the proposal of a 'green belt' is viable in this context (when you clearly expressed the difficulties of finding available land to build green spaces in the city center, where interventions are most needed)? My personal feeling is that maybe some 'urban forestry' approaches would fit better the context of the city center (or the combination of the two approaches), but it is up to the authors to decide, obviously.

Specific comment:

The urban heat island effect discussed in lines 370-377 may be corroborated by citing the study: Zargari, M., A. Mofidi, A. Entezari & M. Baaghideh (2024). Climatic comparison of surface urban heat island using satellite remote sensing in Tehran and suburbs. Scientific Reports 14(1) (Janeiro 5): 643. doi:10.1038/s41598-023-50757-2.

Author Response

  1. The introduction is clear and well framed by the references used. My only comment is that authors should explain better in what way do the chosen factors (especially the socioeconomic and demographic factors) express a measure of "environmental justice", and how is the proposed analytical method capable of expressing its increase/decrease over time.

To address the relationship between the chosen socioeconomic and demographic factors and measures of environmental justice, our study elaborates on how these factors contribute to spatial inequalities in access to green spaces. We acknowledge that environmental justice encompasses not only physical access but also the equitable distribution of ecosystem service benefits across different social and economic groups. Our analysis specifically considers four key factors: population density, land price, the ratio of the population aged under 14, and the ratio of the population aged over 60 to the total population. These factors are critical in understanding the equitable distribution of green space ecosystem services.

  1. Population Density: High population density areas often face challenges in providing adequate green space per capita, potentially leading to unequal access to ecosystem services. Our study assesses how densely populated areas might be underserved in terms of green space availability and quality.
  2. Land Price: This factor is a proxy for economic status and urban development intensity. Higher land prices often correlate with reduced public green space availability, influencing the accessibility and distribution of ecosystem services. Our analysis explores how areas with varying land prices differ in their access to green space benefits.
  3. Ratio of Population Aged Under 14: This demographic indicator helps in assessing green space needs for younger populations, who require recreational and educational ecosystem services. We examine if areas with a higher proportion of young residents have sufficient access to green spaces that cater to their specific needs.
  4. Ratio of Population Aged Over 60: Similarly, the needs of older populations for green spaces differ, often focusing on accessibility and health-related ecosystem services. Our study analyzes whether areas with a higher elderly population have appropriate access to green spaces that support their wellbeing.

There are undoubtedly other relevant factors that could further enrich our understanding of environmental justice, but were not considered in this study due to the lack of available data. Future research could explore these additional factors, provided that more comprehensive data becomes accessible, to paint a fuller picture of the complex dynamics at play in urban green space planning and its impact on environmental justice.

  1. Also regarding the 'Introduction' and the 'Materials and Methods' sections, authors often mention 'green space' or 'urban green space', but what is seen in Figure 1 and throughout all of the study is that they only consider 'public green spaces', especially public parks and gardens. A quick view of Tehran's aerial photographic mosaic will reveal a finer 'mesh' of [private] urban green spaces that are left out of the study, but obviously contribute with ecosystem services and amenities to the city and are, in my perspective, not neglectable. Thus, authors should be clear(er) on how they chose the patches of [public] green spaces, and why was that their choice.

Your observation regarding the study's focus on 'public green spaces' raises an important point. You rightly pointed out that our analysis primarily considers public parks and gardens, as evident in Figure 1 and throughout the study. This approach was not a result of excluding private green spaces initially; rather, it was guided by the selection criteria based on the "Tehran Green Space Master Plan." This plan identifies certain green spaces as vital for the city and its citizens due to their social, recreational, cultural, and environmental significance. These green spaces are generally larger than 11 hectares.

We concur with your perspective that private and smaller green spaces also contribute significantly to the ecosystem services flow, primarily in environmental aspects. However, as these green spaces are not publicly accessible, they lack the broader social aspect of ecosystem service delivery. However, to address this and ensure a more comprehensive analysis, we included "Other green spaces (below 11 hectares)" in Table 1. This factor accounts for smaller green spaces in the study, thereby acknowledging their role in the ecosystem services flow and obtaining the two cost layers.

  1. I am not familiar with the softwares used (neither with the methodology) but but the results and the discussion seem well conducted, except I would also like to see a diagram expressing the age of the different neighbourhoods, because it seems to me that a number of factors (population density, per capitaurban space area, average building height) are highly correlated with the age of the neighbourhoods, within Tehran. There should be a significant amount of literature mentioning that this is a common problem to many 'historical' cities within the Eurasian continent.

Thank you for pointing out the relevance of the correlation between urban factors and the age of neighborhoods in Tehran. While this aspect is indeed significant, we have chosen to maintain our primary focus on urban green space connectivity. However, to address your comment, we have added a brief note in the section discussing Figures 5 and 6. This note highlights the oldest districts of Tehran (11, 12, 10), underscoring the historical development and the imbalance between the growth of built-up areas and green spaces over time. We believe this succinct addition provides necessary context without detracting from our study's main objective.

4.Finally, authors "suggest a 'green belt plan' and the integration of vital ecosystem service corridors into the city's spatial organization". Do you consider that the proposal of a 'green belt' is viable in this context (when you clearly expressed the difficulties of finding available land to build green spaces in the city center, where interventions are most needed)? My personal feeling is that maybe some 'urban forestry' approaches would fit better the context of the city center (or the combination of the two approaches), but it is up to the authors to decide, obviously.

Thank you for your insightful suggestion regarding the feasibility and contextual applicability of the 'green belt plan' within Tehran's urban landscape. Your comment has been instrumental in refining our approach and enhancing the clarity of our proposed strategies. In response to your feedback, we have revised the relevant section of our conclusion.

Specific comment:

5.The urban heat island effect discussed in lines 370-377 may be corroborated by citing the study: Zargari, M., A. Mofidi, A. Entezari & M. Baaghideh (2024). Climatic comparison of surface urban heat island using satellite remote sensing in Tehran and suburbs. Scientific Reports 14(1) (Janeiro 5): 643. doi:10.1038/s41598-023-50757-2.

Thank you for suggesting the Zargari et al. (2024) study. It has been added to the paper to enhance the discussion on urban heat islands.

Reviewer 2 Report

Comments and Suggestions for Authors

The paper presents valuable insights into the various factors affecting the connectivity of urban green spaces, and highlights important points on environmental justice. The authors demonstrate a holistic approach that is valuable for future urban and environmental planning.

The following revisions are recommended:

- There is some repetition of content in the introduction and literature review sections. Please review these to minimise repetition; the two sections could potentially be combined.

- Figure 1: The legend only shows "high" and "low", without clarifying what is being measured. Please amend this to make it clear that it refers to population density.

- Lines 177-190: More details about the Delphi study should be included. For instance, were the experts based in Iran or was this an international group? More information about how the experts were selected would be beneficial.

- Figures 3, 5 & 7: Clarify what "High" and Low" mean, so that readers can quickly understand the figure, without having to look at the text for clarification.

- Lines 466-471: The transition from one sentence to the next one is no clear. Please review for clarity.

- Line 471: Online sources show that Tehran already has a well-developed green belt, please review this sentence to clarify what is being proposed by the authors and how it is a new proposal.

- Line 467: Regeneration of "inefficient" neighbourhoods - please review the use of the term "inefficient" - it is not clear what is meant by this term.

Comments on the Quality of English Language

The paper is clear and well written. The following minor edits are recommended:

- Replace the word "like" for other expressions, for example "such as". I recommend this replacement because "like" is considered informal writing by many academics. See how "like" was used in lines 398, 445 and other parts of the paper.

- Lines 466-471: the writing is confusing, and no capital letter is used after the full stop; it is unclear if the two sentences should have been connected or separated. Please review this paragraph to ensure it is clear for readers.

Author Response

The paper presents valuable insights into the various factors affecting the connectivity of urban green spaces, and highlights important points on environmental justice. The authors demonstrate a holistic approach that is valuable for future urban and environmental planning.

The following revisions are recommended:

  1. There is some repetition of content in the introduction and literature review sections. Please review these to minimise repetition; the two sections could potentially be combined.

Thank you for your suggestion. We have combined the Introduction and Literature Review in the revised manuscript to minimize repetition and enhance clarity.

  1. Figure 1: The legend only shows "high" and "low", without clarifying what is being measured. Please amend this to make it clear that it refers to population density.

We have amended the legend in this figure to explicitly indicate that "High" and "Low" refer to population density.

  1. Lines 177-190: More details about the Delphi study should be included. For instance, were the experts based in Iran or was this an international group? More information about how the experts were selected would be beneficial.

Thank you for your comment requesting more details about the Delphi study. In response, we have expanded the relevant section in the manuscript to include additional information. This includes the process of identifying 45 experts based on their expertise and knowledge of Tehran, the participation of 35 experts who formed an international panel, and their diverse professional backgrounds. This also includes a description of the iterative process used, where initial weights obtained from the experts were sent back for a second review to refine and achieve consensus. These details should provide a clearer understanding of the methodology and composition of our expert panel.

  1. Figures 3, 5 & 7: Clarify what "High" and Low" mean, so that readers can quickly understand the figure, without having to look at the text for clarification.

We have clarified the meanings of "High" and "Low" in the legends of Figures 3, 5, and 7 as suggested.

  1. Lines 466-471: The transition from one sentence to the next one is no clear. Please review for clarity.

Your feedback regarding the clarity and punctuation in lines 466-471 is greatly appreciated. In response, we have revised and rewritten this paragraph to improve its readability and ensure grammatical correctness.

  1. Line 471: Online sources show that Tehran already has a well-developed green belt, please review this sentence to clarify what is being proposed by the authors and how it is a new proposal.

Thank you for your insightful suggestion regarding the 'green belt plan' within Tehran's urban landscape. Your comment has been instrumental in refining our approach and enhancing the clarity of our proposed strategies. In response to your feedback, we have revised the relevant section of our conclusion.

  1. Line 467: Regeneration of "inefficient" neighbourhoods - please review the use of the term "inefficient" - it is not clear what is meant by this term.

Thank you for pointing out the ambiguity of "inefficient" in line 467. We have replaced it with "multifacetedly challenged" to more accurately describe the neighborhoods' complex issues.

Comments on the Quality of English Language

The paper is clear and well written. The following minor edits are recommended:

  1. Replace the word "like" for other expressions, for example "such as". I recommend this replacement because "like" is considered informal writing by many academics. See how "like" was used in lines 398, 445 and other parts of the paper.

Thank you for your suggestion to replace "like" with "such as" for formality. All instances in the paper, including lines 398 and 445, have been revised accordingly.

  1. Lines 466-471: the writing is confusing, and no capital letter is used after the full stop; it is unclear if the two sentences should have been connected or separated. Please review this paragraph to ensure it is clear for readers.

Your feedback regarding the clarity and punctuation in lines 466-471 is greatly appreciated. In response, we have revised and rewritten this paragraph to improve its readability and ensure grammatical correctness.

Reviewer 3 Report

Comments and Suggestions for Authors

I have given this paper an overall merit of High because I consider it to be well-researched, particularly in terms of the methods, and well presented.  However, the paper needs to explain 'ecosystem services' fully [eg lines 261-2] so that the results can be discussed more fully. For example, lines 276-284, explain the pertinent 'social dimensions' but not 'ecosystem services'. Likewise,  the difference between 'ecosystem service hub' and 'corridors' needs to be explained.  This will allow a more detailed discussion of the results, including the particularities of 'environmental justice' in the context of Tehran.  This will allow for a more expanded conclusion. 

Author Response

I have given this paper an overall merit of High because I consider it to be well-researched, particularly in terms of the methods, and well presented.  However, the paper needs to explain 'ecosystem services' fully [eg lines 261-2] so that the results can be discussed more fully. For example, lines 276-284, explain the pertinent 'social dimensions' but not 'ecosystem services'. Likewise, the difference between 'ecosystem service hub' and 'corridors' needs to be explained.  This will allow a more detailed discussion of the results, including the particularities of 'environmental justice' in the context of Tehran.  This will allow for a more expanded conclusion. 

Thank you for your valuable feedback and high appraisal of our paper. We have addressed your suggestions in the revised manuscript. A comprehensive definition of “ecosystem services” relevant to our study's context has been added to the introduction section. This aims to provide a clearer framework for discussing the results, particularly in terms of environmental justice in Tehran.

Additionally, in Section 4.1 (lines 276-284 of the revised manuscript), we have included an explanation emphasizing that ecosystem services encompass social dimensions. This elaboration clarifies the interplay between ecosystem services and social aspects, and how these influence and are influenced by the demand for ecosystem services flow.

To further clarify, we've defined 'green space hubs' as significant areas in an urban environment providing a variety of ecosystem services, characterized by their size and role in various aspects including social and environmental. 'Corridors', on the other hand, are connections between these hubs, facilitating movement and service flow. These additions will enable a more detailed discussion of the results and an expanded conclusion.
